# Comparison of Acoustic Streaming Flow Patterns Induced by Solid, Liquid and Gas Obstructions

**DOI:** 10.3390/mi11100891

**Published:** 2020-09-26

**Authors:** Hsin-Fu Lu, Wei-Hsin Tien

**Affiliations:** Department of Mechanical Engineering, National Taiwan University of Science and Technology, Taipei 10607, Taiwan; frank0975167008@gmail.com

**Keywords:** acoustic streaming, steady streaming, particle tracking velocimetry, acoustofluidics

## Abstract

In this study, acoustic streaming flows inside micro-channels induced by three different types of obstruction—gaseous bubble, liquid droplet and solid bulge—are compared and investigated experimentally by particle tracking velocimetry (PTV) and numerically using the finite element method (FEM). The micro-channels are made by poly(dimethylsiloxane) (PDMS) using soft lithography with low-cost micro-machined mold. The characteristic dimensions of the media are 0.2 mm in diameter, and the oscillation generated by piezoelectric actuators has frequency of 12 kHz and input voltages of 40 V. The experimental results show that in all three obstruction types, a pair of counter-rotating vortical patterns were observed around the semi-circular obstructions. The gaseous bubble creates the strongest vortical streaming flow, which can reach a maximum of 21 mm/s, and the largest u component happens at Y/D = 0. The solid case is the weakest of the three, which can only reach 2 mm/s. The liquid droplet has the largest *v* components and speed at Y/D = 0.5 and Y/D = 0.6. Because of the higher density and incompressibility of liquid droplet compared to the gaseous bubble, the liquid droplet obstruction transfers the oscillation of the piezo plate most efficiently, and the induced streaming flow region and average speed are both the largest of the three. An investigation using numerical simulation shows that the differing interfacial conditions between the varying types of obstruction boundaries to the fluid may be the key factor to these differences. These results suggest that it might be more energy-efficient to design an acoustofluidic device using a liquid droplet obstruction to induce the stronger streaming flow.

## 1. Introduction

Acoustic streaming is a steady flow phenomenon due to the interaction between flow and boundaries under high frequency oscillation. In a mm-size or smaller closed space, the strength of acoustic streaming flow is strong enough for applications such as trapping/manipulation of particles [1,2,3,4], flow mixing and flow control [4,5,6,7]. The exciting oscillation frequency can be to the order of MHz and induced by surface acoustic wave (SAW) in a microfluidic device [8,9,10,11]. In other studies, the exciting oscillation frequency is in the range near audible frequency range (~kHz) and the streaming flow is often excited with certain type of obstructions are present to enhance the streaming flow. The steady flow patterns are sometimes also referred as “steady streaming” [1,2]. This mechanism to drive the microscale flow is simple to implement, and does not require high pressure gradient, which reduces the difficulty of sealing for microfluidic device. Compared to other driving flow or particle driving mechanisms such as electrokinetics, electrophoresis or optical tweezers, the excitation mechanism is purely mechanical and without any factors that may introduce chemical reactions. Therefore it is considered to be more biocompatible and less sensitive to the composition of the working fluid [12]. 

The theory behind acoustic streaming has been developed since it was first described by Rayleigh in 1883. Summaries of classical theories and pioneer developments can be found in earlier reviews [13,14]. A more recent review by Wu [15] emphasizes the applications in biomedical research and sonoporation. Recent development of the microfluidics emphasizes the vortical streaming flows induced outside of the Stokes layer, namely the Rayleigh streaming, with the theory developed based on the interaction between the Stokes drag and acoustic radiation force on the particles in the flow [16,17,18,19,20,21]. Analytical solutions can be developed for limited cases, and simulations based on perturbation theory are also developed [22]. The details of the acoustic streaming flows are still not fully understood, partly because of the wide range of temporal and spatial scale of the flow. Near the channel side wall inside the boundary layer, the flow has a time scale in the order of microseconds due to the kHz to MHz frequency of oscillation. The Stokes boundary layer has the length scale of $\delta \sim \sqrt{\nu /\omega }$, where ν is the viscosity kinematic viscosity and ω is the angular frequency of the oscillation. In microfluidics δ is on the order of a few micrometers. On the other hand, the induced streaming flow outside of the Stokes layer is steady, the time scale can be considered in seconds. The length scale of the vortical streaming flow is related to the size of the obstruction or the channel dimension, which is a few hundreds of micrometers at most. Therefore, most of the studies are done experimentally and the popular measurement methods are flow visualization or PIV (particle image velocimetry)/PTV (particle tracking velocimetry). 

One of the commonly used geometries to enhance local acoustic streaming flows is a bulged geometry at the sidewall of the channel. The obstructions are either solid shape bulged out of the channel sidewall [1,2,5,23,24,25] or gaseous bubbles designed to be trapped at specific locations in the channel [3,6,7,26,27,28,29]. The excited streaming flow patterns are similar, but there are two notable differences in the two types of obstructions other than the similar geometry. The first difference is whether the incompressibility condition is satisfied or not. The solid bulge is incompressible, and the gaseous bubbles are apparently compressible. The other difference is the flow boundary conditions, namely slip/no slip conditions at the flow boundary. With the solid bulge, the boundary is a solid-fluid interface and it should be considered as no-slip boundary condition. On the other hand, for the gaseous bubble obstructions, the interface is made by a liquid–gas boundary and should be considered as a slip boundary. A preliminary investigation in the previous study compared the streaming flow patterns induced by a gaseous bubble and a semicircular obstruction at the microchannel wall using 3-D Micro PTV [30]. Both the gaseous bubbles and solid bulges with diameters of 0.2 to 0.5 mm diameter were tested. The experimental results show that the gaseous bubble obstruction has a better ability in converting oscillatory kinetic energy into the fluid to form steady-streaming vortices. Stronger steady-streaming vortices is formed by the gaseous bubble, and the vortices generated are found to have the highest velocity in the core region of the vortices. However, the question still remains unclear why the streaming flow is influenced by the gaseous bubbles more, and which of the two differences mentioned plays the major role in enhancing the streaming flow patterns. This question can be answered by introducing the third phase of the physical matter: a liquid droplet as a third kind of obstruction. A liquid droplet obstruction is incompressible yet in fluid phase, which will make the interface between the working fluid and the obstruction to be a liqui–liquid interface, which is a slip boundary. To the best of the authors’ knowledge, there are no previous studies focus on the comparison of the acoustic streaming flows of different types of obstruction, either experimentally or computationally. Based on these aforementioned works and deduction, this study is thus intended to compare the different flow patterns induced by three types of obstructions: solid bulge, liquid droplet and gaseous bubble. For the first time, microfluidics designed to have the three types of obstructions of the same characteristic length are manufactured and measured with micro-PTV to acquire the detailed velocity and vorticity fields. These experimental data are used to determine and explain which of the three obstruction types produces the strongest streaming flow. 

The organization of this study is explained as follows. In the following sections, the experimental setup is explained first, followed by the experimental results, discussions and the limitations of the study, and finally the conclusions can be made from these results.

## 2. Materials and Methods 

The microchannel configurations used in this study and the process flow for making them are shown in Figure 1. Figure 1a shows the semi-circular bulge in the middle of the channel to induce the acoustic streaming flow. In Figure 1b, the semi-circular bulge is replaced by a microchannel passage to allow the liquid droplet or gaseous bubble to get into the main channel. The amount of gas or liquid is controlled manually to keep the top view of the extruded bubble or droplet to remain in semi-circular shape. In this study, the gaseous bubble is created by filling air into the passage, and for the liquid droplet mineral oil is chosen for filling the passage and to form the liquid droplet. The width of the main channel is 2 mm, and the height of the channel is 250 μm. The semi-circular solid bulge has a diameter of 0.2 mm and the width of the flow passage for introducing the liquid droplet and gaseous bubble is also set to 0.2 mm in order to form the same geometry to the solid bulge. The dimension of the study was chosen to meet the precision limitation of the micromachining for the fabrication of the master mold, which is ±5 μm (0.005 mm). The width and the height of the channels were chosen in order to observe the localized streaming effect generated by the obstruction geometry. A survey of the previously published results show that the size is larger but comparable to other works [1,2,4,5,6,7,8,23,26,29], so the difference in size should be of the similar order and the results should be still in a comparable range. 

The obstruction radius for the gaseous bubble and liquid droplet were known to deform due to the oscillation in acoustic streaming flows. There are previous studies that show the oscillation patterns of the bubble could change the acoustic streaming flow patterns [4,27], and the flow patterns were dominated by oscillation frequency. In this study, since the oscillation frequency is fixed, the change of size and volume of the fluid obstructions (bubble and droplet) should be less than 5% [4] and considered as a fixed size [27] according to the previous studies. The precision error due to machining can be quantified as ±5 μm. In addition, the size of the fluid obstructions was controlled by the control device to prevent gradual volumetric changes due to pressure variations. As shown in Figure 2, the changes in volume of the liquid droplet and gaseous bubble during the experiments were maintained at minimum with a manual control through a syringe. The piston of the syringe is attached to a set screw, which can adjust the volume of the syringe with fine increments by the thread pitch. It was found that the volume changes throughout the experiments is within 5% with the controlling mechanism, and the time scale for a significant change of volume (larger than 5%) without the control is of the order of 10 s. Therefore, the manual control mechanism is capable of compensating the volumetric changes and the change in gaseous bubble or liquid droplet volume is not a significant factor to the experimental errors.

The microchannel is made by a standard micro-molding process, and the master mold is made by low-cost micro-milling. The two configurations were made by two slightly different processes. Figure 1c shows the process flow of the configuration of the solid semi-circular bulge as shown in Figure 1a. The solid bulge utilizes PMMA (polymethylmethacrylate) as the base material and the master mold was milled to achieve the required geometry. PDMS (polydimethylsiloxane) was used to make the channel side of the microchannel by rolling over the PMMA master mold. The configuration in Figure 1c was made by a two-step rollover process shown in Figure 1d. In the first step, a PMMA negative mold was made by micro-milling to create the channel geometry to avoid unwanted rounded corner caused by the milling tool. The second step is to create a positive EPOXY mold for a second rollover to create smooth channel floor for the transmitted light microscope imaging. A PDMS rollover made from the PMMA negative mold serves as an intermediate positive mold. By attaching a glass slide on top, the PDMS poured into the void space in between can form the hollow mold of the microchannel side wall. After baking, the hardened hollow PDMS part was bonded to another smooth PDMS base plate by plasma bonding to create a negative mold for making the EPOXY positive mold. With the EPOXY mold, the side wall height can be precisely controlled, and a smooth channel floor surface free of milling tool mark can be created. The PDMS channel can be made from the second rollover from the EPOXY positive mold. To complete the making of these PDMS channels, they were bonded with the glass slide by oxygen plasma bonding. 

The acoustic streaming flow was induced by the oscillation of a pair of double-layer ceramic piezoelectric disks (T216-A4NO-273X, Piezo Systems, Woburn, USA). The piezoelectric disks were driven by a 12 kHz oscillation signal generated by a signal generator (HDG2022B, Hantek, Qingdao, China) and high-speed bipolar amplifier (HSA 4012, NF Corporation, Yokohama, Japan) with $V_{\mathrm{PP}}$ set to 40 V. The piezoelectric transducers were placed on the microchannel and the positions are shown in Figure 1a,b. To effectively transfer the oscillation of the transducers into the microchannel, the transducers were chosen to be placed as close as possible to the obstruction without blocking the light path. In the current study, polystyrene particles of 2.6 μm diameter (PP-35-10, Spherotech, SG = 1.05) were used due to its density close to water. The Stokes number for the experimental conditions is calculated to be $S_{\mathrm{tk}}=8\times 10^{-4}\ll 1$, which ensures the validity of the PTV technique. The working fluid is water and particles were pre-mixed into suspension solution on a mass fraction of 0.1%.

The experimental setup is shown in Figure 3. The system was built around an inverted microscope. The microchannel with piezoelectric disks was located on top of the platform. The high frequency oscillation was first generated by the signal generator and then amplified by the signal amplifier to the required oscillation frequency and voltage. The actuated flow patterns and detailed PTV analysis was recorded by a monochrome PIV camera (1392 pixel × 1040 pixel, pixelfly, PCO) and illuminated by a pulsed LED light source. The magnification in this study is fixed to 10× in order to capture the full range of the streaming flow. The synchronization of the camera and the pulsed LED was done in the frame straddling mode and the exposure time and time interval were set to 1.5 and 0.9~1.5 ms, respectively. The resulting data acquisition rate is 6.75 image pair/s.

The acquired images were illuminated by transmitted lights and thus particle images are the dark shadows with the bright background. This has to be inverted so that the peak-finding algorithm in PTV processing can be used to find the particle image centers. There are also the background images of obstruction and the channel wall, which cannot be processed during the tracking algorithm and need to be removed. These processing steps are shown in Figure 4 and works as the pre-processor for the PTV data processing. Due to the camera’s special PIV mode, each image pair has intensity difference in between the two frames and requires separate average and background removal to achieve better image quality. 

The PTV processing algorithm in this study is based on the work of Lei et al. [31]. An in-house GUI interface written in MATLAB was developed to process the image pairs. Due to the limited particle images in each image pair, each velocity field result was obtained by overlapping all the found PTV vectors extracted from one video sequence of single experiment, and the erroneous vectors was filtered by the PTV version of universal outlier method proposed by Duncan et al. [32] and masking of the obstruction and background. The vector field was interpolated onto a uniform grid for better understanding of the physics.

To help understanding the physics behind the observed flow patterns, additional numerical simulations has been performed using finite element method (FEM) with multi-physics. The numerical model is adopted and adjusted from the method proposed by Muller et al. [22] for the microfluidic device in the current study. The geometry for the numerical model is idealized as a 2-D rectangular domain of 15 mm long and 2 mm wide to simulate the actual device. A semi-circular obstruction is set in the bottom wall at the center of the channel. These dimensions were set to match the experimental setup. 

To best simulate the role of the obstruction, several boundary conditions were tested to match the flow patterns observed in the actual experiments. For the solid bulge case, the oscillation of the piezo plate is transferred into the numerical model through the oscillating velocity boundary conditions along the solid bulge and/or the entire solid boundaries:(1)$$u_{0}=U_{0}e^{-i\omega t}\;or\;v_{0}=V_{0}e^{-i\omega t}$$
$U_{0}$ and $V_{0}$ are constant, ω=2πf is the angular frequency of the oscillation. The oscillation direction was set to be either in the horizontal (X) or vertical (Y) directions. For fluid type obstructions, the oscillation velocity boundary condition is set to match the shear-free fluid–fluid interfacial boundary condition, which assume that the semi-circular obstruction has an oscillatory velocity component only in the radial direction:(2)$$u_{0}=U_{0}\left(\frac{x}{R}\right)e^{-i\omega t}\;and\;v_{0}=V_{0}\left(\frac{\sqrt{\left(R^{2}-x^{2}\right)}}{R}\right)e^{-i\omega t}$$
R is the radius of the semi-circular obstruction and x is the x-coordinate along the channel wall. 

Due to the wide range of the time scale in these kind of problems, the numerical solution is obtained from a two-step solution procedure. Outside the viscous boundary layer, the time-averaged streaming flow can be considered as incompressible as long as the acoustic wave-length is much larger than the scale on which the physics occurs [14], and therefore it is the differences caused by the obstruction types inside of the viscous boundary layer that could contribute to the results. Inside the boundary layer, the flow can be described by thermoacoustic equations [22] and the differences in obstruction only comes into play through boundary conditions, which is the oscillation velocity of the wall induced by the piezoelectric oscillations. The inner solution (inside the boundary layer) was first obtained and the results were fed to the solver to solve the outer solution.

## 3. Results

The results of PTV processing of the three different obstruction types are shown in Figure 5a–f respectively in both velocity vector plots with speed (absolute velocity magnitude, absV) and with vorticity contour maps. The reference vector shown on the top right is 2 mm/s. It can be observed that in all the streaming flow patterns there are two counter rotating vortices located around the hemispherical obstructions. The shape of the counter-rotating streaming vortices is more circular for the solid bulge than the other two types of obstructions. The absV maps also show that the vortical structures have higher speed near the obstruction surface but lower speed away from the obstruction, which suggests that the fluid is accelerating when it is moving towards the obstruction surface. The rotating directions of the flow are not the same for the three phases; for the solid bulge case, the vortices entrain the flow from the dome of the hemisphere and reject the fluid outward from the side of the hemisphere. For the liquid droplet and gaseous bubble obstructions, the rotation direction of the vortices is reversed, bring the fluid inward from the side of the hemisphere and reject the fluid from the top of the hemisphere. From the vorticity plot, it can be observed that for gaseous bubble obstruction case the locations of the maximum vorticity are nearest to the sidewall of the channel, and the solid bulge are at the farthest to the wall. In general, the gaseous bubble obstruction creates the strongest flow while the vortical flow pattern of the solid bulge obstruction is the weakest of the three, and the flow pattern created by the liquid-type obstruction is weaker than the gaseous bubble case but significantly stronger than the solid bulge. The maximum level of speed for the solid bulge obstruction is 3 mm/s, much lower than 20 mm/s for the gaseous bubble and 12 mm/s for the liquid type. The vorticity maps show that the center of the streaming vortex pair is closer to the top of the solid bulge obstruction than the other two types of obstruction, which both located closer to the side of the obstructions.

To further analyze and compare the flow patterns of the three types of obstruction, the extracted line profiles of the velocity components U, V in the X, Y directions, and the absolute velocity absV $=\sqrt{U^{2}+V^{2}}$ are plotted along the selected locations. Figure 6 shows these profiles of the solid bulges. The positive and negative values of the u and v components are mainly due to the directions of the two counter-rotating streaming vortices around the solid bulge. From Figure 6a the peak magnitudes of u and v components are at X/D = −0.8, XD = −1.0 on the left side of the obstruction (X/D < 0). On the right side of the obstruction (X/D > 0), the peak values are not as large for both u and v components. This is due to the asymmetric counter rotating vortices shown in the vector plot. Due to the same reason, in Figure 6b,c the velocity profiles are not the same. These results may come from the imperfect shape of the obstruction or the slightly shifted locations of the two piezo plates. A common feature can be observed is that the two velocity peaks at different Y/D locations are not comparable as well; The peak of both u and v components are larger closer to the obstruction, and the peak at the outer region away from the obstruction decays with a tail extended to far-field. This feature is because the streaming vortices is located the side of the obstruction but the streaming flow affects the far-field away from the obstruction. This also suggests that the fluid in the streaming vortex has to accelerate while approaching and decelerate while leaving the obstruction. This is more apparent in the v component shown in Figure 6d, as at the centerline of the obstruction the surface at X/D = 0 and Y/D = 0.5 is the stagnation point and the v component accelerates quickly to a maximum value and decay to the far-field away from the obstruction, while the u component has both positive and negative values. This is because of the asymmetric patterns of the two streaming vortices. 

Figure 7 shows the velocity and speed profiles at the selected locations for the case of the liquid droplet obstructions. The u and v components in Figure 7a shows that the rotation directions of the counter-rotating streaming vortices are inverted compared to the solid bulge case. The fact that the u component on the right side of the obstruction is mostly negative and positive on the left side suggests that the center of the rotation is located lower than Y/D = 0.57. This is also different compared to the solid bulge case shown previously. In The v component in Figure 7b,c have negative peaks around Y/D = 0.4 and very low positive values on the far-side away from the obstruction. This is mainly because the streaming vortex pairs are located away from the side of the droplet obstruction, and these two locations (X/D = −0.56 and X/D = 0.53) are already close to the edge of the vortex, therefore the flow direction is mainly going down. On the other hand, the u components shown in these two figures have both positive and negative peaks. The locations of the peaks roughly indicate the regime of the streaming vortices. The decay of both of the u and v components to the far-field is similar to the previous case and indicates the fact that the streaming velocity is higher when approaching the liquid droplet obstruction and lower when moving away. Figure 7d shows the velocity profile at the centerline of the liquid droplet. Compared to the case of the solid bulge, the trend of the v component is similar but opposite in direction, which quickly increases from zero to a positive maximum and then decreases gradually as further away from the obstruction surface. The u component varies very little and is close to zero. This is because the two streaming vortices are more symmetric, so the net flow in the X direction is roughly zero. 

Figure 8 shows these profiles of the gaseous bubble obstructions. Compared to the previous two cases, the velocity profile for the gaseous bubble case in Figure 8a shows more similarity to the liquid droplet and different from the solid bulge. This is as expected since the flow direction of the counter-rotating vortex pair of the gaseous bubble case is the same as the liquid droplet and opposite to the solid bulge case. In Figure 8b,c, the values of the u and v component on both sides of the obstruction (X/D = −0.47 and X/D = 0.48) have positive and negative peaks. This is different than either the case of the solid or liquid droplet obstruction cases. This may be due to two reasons. First, the locations of the pair of the streaming vortices are lower than the liquid droplet case and also closer to the obstruction’s surface, so the profile at the selected locations include the streamlines approaching and leaving the obstruction. Second, the opposite direction of rotation to the flow patterns for solid case results in the inverted profiles for the u and v components. The profiles shown in Figure 8d is just similar to the liquid droplet case and has an inverted pattern if compared to the solid bulge case. This is also as expected since the streaming flow direction of the gaseous bubble case is the same as the liquid droplet case.

## 4. Discussion

Figure 9, Figure 10 and Figure 11 are the comparison of the horizontal u, v components and speed profiles of the three cases at Y/D = 0, Y/D = 0.5 And Y/D = 0.6. The locations are selected for best illustrating the differences between the three cases. It can be observed that due to the opposite flow directions, the solid bulge case shows roughly inverted profiles compared with the other two cases. The comparison on the magnitudes shows that the solid bulge case indeed has the lowest velocity and speed, indicating the weakest streaming flow of the three. The air bubble case has the largest u component at Y/D = 0, and the liquid droplet has the largest v components and speed at Y/D = 0.5 and Y/D = 0.6. With the same configuration and experimental conditions, it can be concluded that the solid bulge performs the worst among the three obstruction types, and the induced streaming flow is the slowest and the circulation regime is smallest. 

Figure 12 shows the comparison of the Y-component velocities for the three obstruction types at X/D = 0. Along the centerline of the semicircle, a decaying Y-component distribution can be observed, and the negative values of the solid bulge is due to the inverted flow pattern compared to the other two fluid obstructions. The highest v-component of the three obstruction cases is the liquid obstruction along X/D = 0. This is probably because the vortical streaming flow patterns are closer to the centerline for the liquid droplet obstruction case. These velocity profiles are similar to the results obtained from the experiments and simulations of the acoustic streaming flow around sharp edges [33], where the geometries might differ from the current study but shares similar physics. Figure 13 and Figure 14 show the comparison of the u-component velocities between the three obstruction types along X/D = −0.6 and 0.6, respectively. The high and narrow velocity values between Y/D = 0~0.3 for the fluid obstruction cases indicate that the flows towards the obstruction near the center of the streaming vortical pattern were accelerating and concentrated. Away from the obstruction at Y/D > 0.6, the negative outward flow velocities were much lesser and extended to a wider span. For the solid bulge, the signs of the flow velocity are inverted because of the inverted flow patterns, and the magnitudes are also low compared to the fluid obstruction cases. This result is mainly because the streaming flow induced by solid bulge is weak, and also partly because the locations of the center of the vortical flow pattern are farer from the centerline.

A more detailed investigation of the induced streaming flow fields of the three obstruction types show that the liquid obstruction is actually transferring the oscillation of the piezoelectric plate most efficiently, creating the strongest streaming flow of the three cases. This is shown in Figure 15, Figure 16 and Figure 17. Figure 15 shows the absolute velocity (speed) averaged over Y/D. It can be observed that the liquid droplet case has the largest averaged speed almost in all Y/D, except a small region close to Y/D = 0. More detailed information near the velocity peak location of the gaseous bubble at Y/D = 0.08 and the velocity peak location of the liquid droplet at Y/D = 0.66 are shown in Figure 16 and Figure 17, respectively. The curve-fitted streaming velocity profiles in these two figures can be compared at the locations of the maximum speed for the liquid droplet and gaseous bubble cases. For the gaseous bubble case, the maximum speed occurs at the outward streaming flow at Y/D = 0.08, and the comparison of the two cases show that although the speed is high, the streaming profile is narrow at this location. For liquid droplet case, the speed reaches maximum at Y/D = 0.66, and the profile is broader and higher than the gaseous bubble case. As a result, the induced streaming flow region is generally larger, and the average speed of the liquid droplet case is also larger.

These comparisons of the three obstruction types reveal two interesting points. First, the streaming flow directions of the gaseous bubble and liquid droplet are opposite to the solid bulge case. For the two fluid obstruction cases, the direction of the circulating flow pattern is outward-going and away from the centerline of the semicircular obstruction surface, and inward-going towards the sides of the obstruction. This is opposite to the case of the solid bulge. These results of the gaseous bubble and liquid droplet cases agree with the previous published results [4,6,7], but the solid case differs from previously published results [5,23]. From the results of the cylinder case in [4], there are four symmetric counter rotating vortices present around the cylindrical obstruction, therefore it is possible to have either of the flow directions at the obstruction. There are several possible reasons for the different directions, but the dominate factor of the circulating flow direction is currently not clear. The second interesting point is that the maximum speed of the streaming flow of the three cases is the one generated by gaseous bubble, which can reach a maximum of 21 mm/s. The maximum speed is for the liquid droplet is seconded to it, reaches 12 mm/s. For the solid case it is significantly slower, only can reach a maximum of 2 mm/s. Both the liquid droplet and solid bulge are both incompressible, thus the compressibility of the material is not likely to be the root cause of this difference, otherwise the difference between the gaseous bubble and liquid droplet obstruction should be larger. The acoustic impedances of water, PDMS and mineral oil have a difference about 40%, which can also be excluded as the main reason of the large difference in streaming velocity between the solid bulge and fluid (gaseous bubble and liquid droplet) obstructions. 

In order to explain the first observation, the FEM simulation were performed and the results are shown in Figure 18. As shown in Figure 18a, the acoustic pressure distributions was obtained from both the solid bulge and fluid obstruction cases. The different distributions came from the different boundary conditions set in Equations (1) and (2), which are due to the results of different interfacial conditions. For the solid bulge, the fluid is oscillation with the wall with no-slip condition applied, but in the case of fluid obstructions, the boundary condition is set by assuming that the semi-circular obstruction has an oscillatory velocity component only in the radial direction to match the shear-free fluid–fluid interfacial boundary condition. 

Figure 18b–d show the comparisons between the numerical and experimental results side-by-side for the three obstruction types respectively. The flow patterns obtained with these different boundary conditions match the experiment results reasonably with a few notable differences. The largest velocity captured in the simulation is located in the region near the obstruction surface about 30 to 70 degree above Y/D = 0 for the solid bulge case, and 45 to 80 degree above Y/D = 0 for the fluid obstruction case. These results are different from the experiments shown on the left side of the subplots in Figure 18b–d. The main reason for this difference is the limitation of the PTV measurement at the near wall region. This region is thin and very close the obstruction surface, and tracer particles have difficulties to flow through and be observed at such near wall region. Therefore, the high-speed region could not be observed experimentally. The other difference between the numerical and experimental results is that the locations of the vertical flow patterns are slightly different. In the experiments, the centers of the vertical flow patterns are closer to the obstruction surface while in the numerical results it is more detached from the surface. The locations of the vortical structures depend on several factors, including geometry, oscillation frequency, working fluid viscosity, acoustic impedance and density. Among these parameters, the acoustic impedance could be the major cause of the differences. This is because the porous structure of the PDMS channel could make the acoustic impedance to deviate from the ideal properties that adopted in the simulation. 

For the solid bulge case, it is found that only when the oscillation was set to the horizontal direction (parallel to the wall) that the flow direction is the opposite to the fluid cases. Otherwise the flow direction is the same. This suggests that for the solid bulge case, the oscillation mode of the combined system of the piezo plates and the PDMS device is important to the acoustic flow patterns.

In Figure 18c,d, the experimental streaming velocity magnitude is significantly smaller for the solid bulge case than the fluid obstruction cases. For the simulation result, the difference is not obvious because the oscillation velocity magnitude used in the current study was set to be the same as the solid bulge case. This may not be the case in the actual case, since the path of transferring the oscillation from the piezo plate to the fluid is different and the efficiency can be very different. Because the fluid obstructions are separated from the PDMS channel by the solid-fluid interface, and can therefore be considered as a separated system with a smaller system mass in parallel to the PDMS microchannel. The oscillation from the piezo plate can be transferred to the fluid in the channel through the fluid obstruction. The solid bulge obstruction, on the other hand, is part of the PDMS microchannel which has a much greater system mass. As a result, the oscillation from the piezo plate has to be distributed to the whole PDMS channel before it can be transferred into the fluids, and therefore can only generate a much smaller oscillation amplitude.

The different results between the droplet and bubble cases could be due to the different properties of the liquid (oil, incompressible, higher density, much higher viscosity) compared to gas (air, compressible, low density, lower viscosity) to make the fluid element to transfer the oscillation of the piezoelectric plate with different efficiency. In this study, both fluid obstructions utilize the same flow passage and are controlled by a closed syringe system. During the experiment, oscillations from the piezo plate causes a squeezing effect to the PDMS flow passage and thus the oscillatory velocity can be considered to be the same for both the liquid and droplet cases. Because of the lower density of gas, the difference in acoustic impedance between the air / water interface is much larger than the oil /water interface, and oscillation wave may get significant reflection and transfer less efficiently. On the other hand, the lower viscosity of gas can reduce the friction damping from the channel wall and the vibrating modes due to the compressibility may compensate for the loss and create the local high-pressure region that raised the local acoustic pressure. These factors may explain the higher overall streaming velocity of the liquid droplet case and the higher peak streaming velocity of the gaseous bubble case observed in this study. 

The locations of the maximum absolute velocity magnitude (absV) are related closely to the flow directions. For the gaseous bubble and liquid droplet cases, the maximum values are located at the sides of the obstruction, while for the solid case it is located at the top of the obstruction. In all three cases the maximum locations are at where the fluid is moving towards the obstruction. This observation suggests that the streaming flow gains kinetic energy when the fluid is approaching the obstruction and slows down at the far-field. This observation can be explained from the study by Mikhail et al. of the acoustic streaming around sharp edges [33]. The streaming pattern with high speed near the obstruction is due to the fact that near the boundary layer, the fluid enters the acoustic force area along the side of the obstruction and is ejected at the tip driven by the centrifugal force.

## 5. Limitation of the Study

Due to the limitation of the hardware, there are a few limitations in the current study. Observations of the raw images suggest that there may be an underestimation of the velocity values for the air bubble case. The time interval Δt is not short enough for the PTV setup to capture the displacement of the particles close to the obstruction. The exposure time for each image could also be too long and the particle image may be elongated to non-circular shape, which can cause the PTV algorithm fails to identify the particle image center, and the displacement vectors therefore may not be resolved correctly. In addition, because of the size of the microfluidic device and limitation of the measurement techniques, it is difficult to measure the acoustic pressure distribution with satisfactory temporal resolution. Because of the limitation of the camera frame rate, the dynamics of bubble and droplet were not investigated in this study. There are also studies indicates the three dimensionality of the streaming flow [28,30] may be important for droplet and bubble obstructions, but it requires different experimental setup and could be further studied in the future. For the numerical simulation, it is difficult to distinguish between gaseous bubble and liquid droplet with the current numerical model. To investigate the difference of gaseous bubble and the liquid droplet numerically, a new numerical model that can include the fluid properties and correctly predict the dynamics of the bubble/droplet oscillation is required. More detailed investigation both experimentally and numerically should be conducted on this topic. 

## 6. Conclusions

In this study, microfluidic devices have a solid bulge, liquid droplet and gaseous bubble type semi-circular obstructions were successfully built by soft lithography using low-cost micro-machined mold. Experimental results with PTV reveal that in all three obstruction types, a pair of counter-rotating vortical patterns were observed around the semi-circular obstructions. The velocity distributions show that the fluid is accelerating when it is near the obstruction surface and decelerates when it is away from the obstruction. This is because the fluid enters the acoustic force area along the side of the obstruction and is ejected at the tip driven by the centrifugal force.

The flow patterns of the gaseous bubble and liquid droplet obstructions show fountain-like flow patterns, which were opposite to the case of the solid bulge. Numerical simulations performed using FEM further show that the interfacial conditions between the different types of obstruction boundary to the fluid causes the opposite vertical flow patterns, and the flow patterns matches experimental results qualitatively.

Comparisons of the profile of velocity components and magnitudes confirm that the solid bulge case has the lowest velocity and speed, indicating the weakest streaming flow of the three. The strongest streaming flow of the three obstruction types is the one generated by gaseous bubble, which can reach a maximum of 21 mm/s, and the largest u component happens at Y/D = 0. The solid case is the weakest of the three, only can reach 2 mm/s. The liquid droplet has the largest v components and speed at Y/D = 0.5 and Y/D = 0.6. 

The induced streaming flow region and average speed are both the largest of the three for the liquid droplet obstruction. This is because of the higher density and incompressibility of liquid droplet compared to gaseous bubble. Therefore, it might be more energy-efficient to design an acoustofluidic device using a liquid droplet obstruction to induce the stronger streaming flow.

## Figures and Tables

**Figure 1 micromachines-11-00891-f001:**
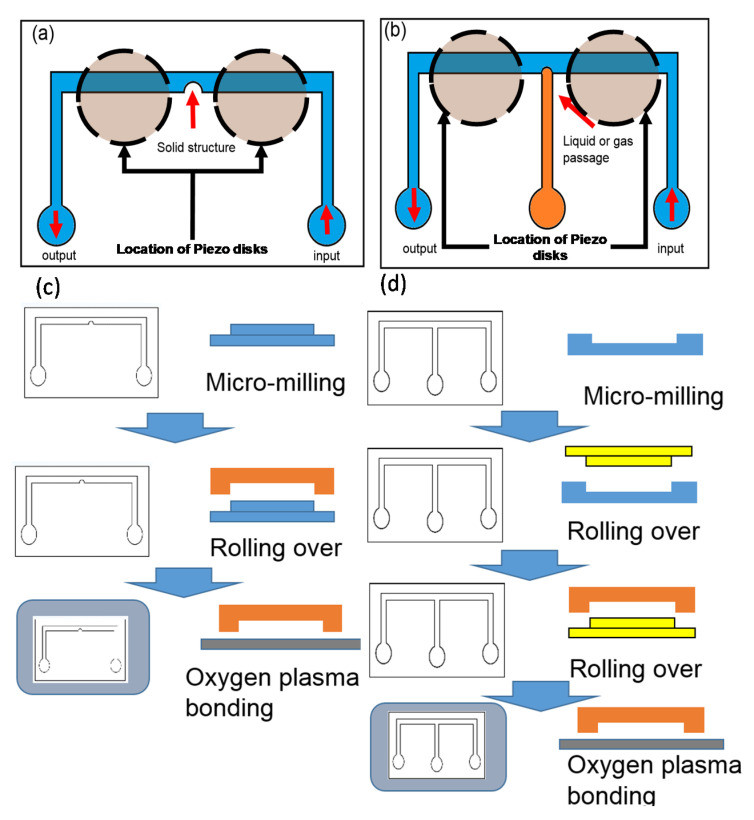
Manufacturing of the microchannel with obstructions to induce acoustic streaming in the current study: (**a**) design for the solid bulge obstruction; (**b**) design for the liquid droplet or gaseous bubble obstructions; (**c**) manufacturing process flow of design (**a**); (**d**) manufacturing process flow of design (**b**).

**Figure 2 micromachines-11-00891-f002:**
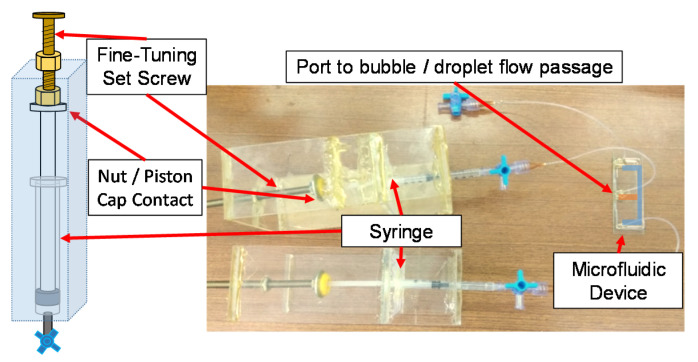
Bubble and droplet control mechanism. The piston cap of the syringe is in contact with the nut sat on the screw. The volume inside the bubble/droplet flow passage can be fine-tuned by controlling the engage/disengage of the nut.

**Figure 3 micromachines-11-00891-f003:**
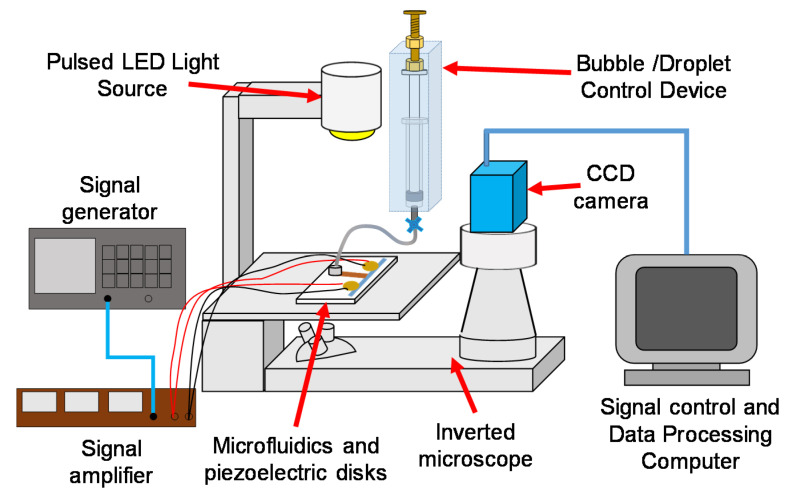
Experimental setup for the flow visualization and particle tracking velocimetry (PTV) measurement.

**Figure 4 micromachines-11-00891-f004:**
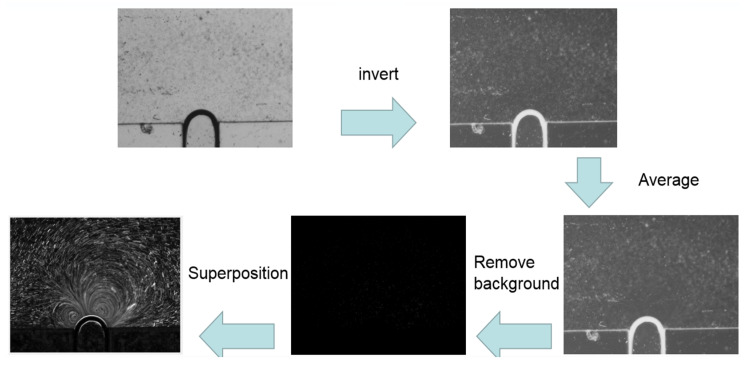
Process flow of image pre-processing to invert the image and remove background.

**Figure 5 micromachines-11-00891-f005:**
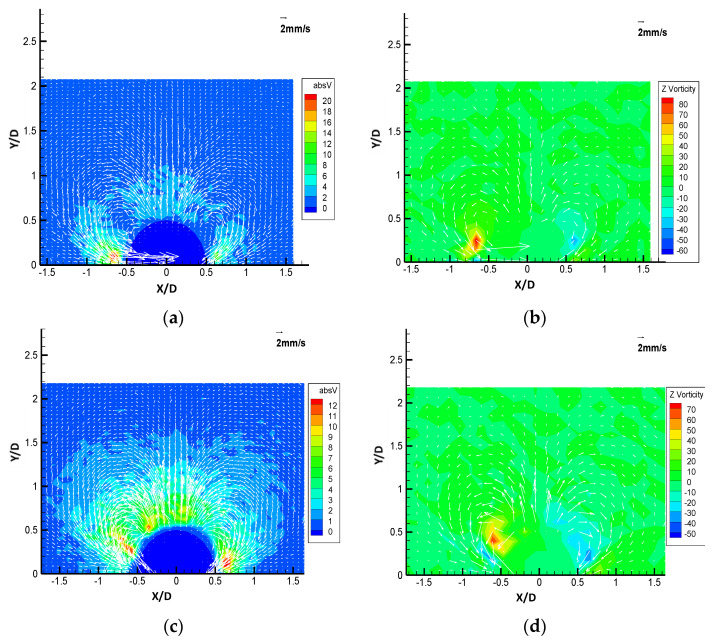
The velocity vector plots with the absolute velocity magnitude (**a**,**c**,**e**) and Z vorticity contour maps (**b**,**d**,**f**) of the three types of obstructions: Gaseous bubble (**a**,**b**), Liquid droplet (**c**,**d**) and solid boundary (**e**,**f**).

**Figure 6 micromachines-11-00891-f006:**
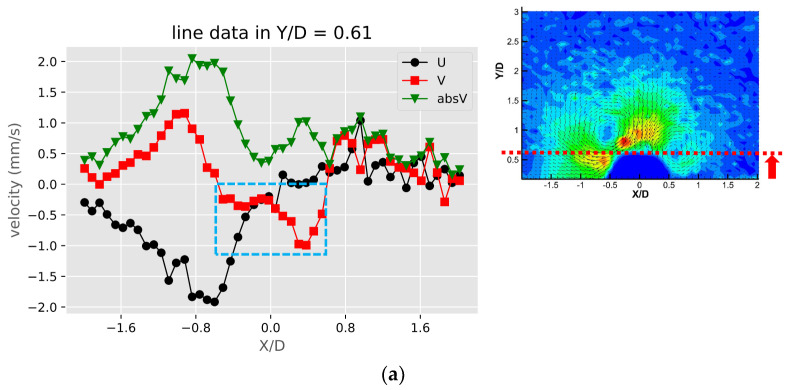
The profiles of velocity components in the X and Y directions (U, V) and absolute velocity (absV) of the solid bulges: (**a**) Y/D = 0.61; (**b**) X/D = −0.51; (**c**) X/D = 0.55 and (**d**) X/D = −0.02.

**Figure 7 micromachines-11-00891-f007:**
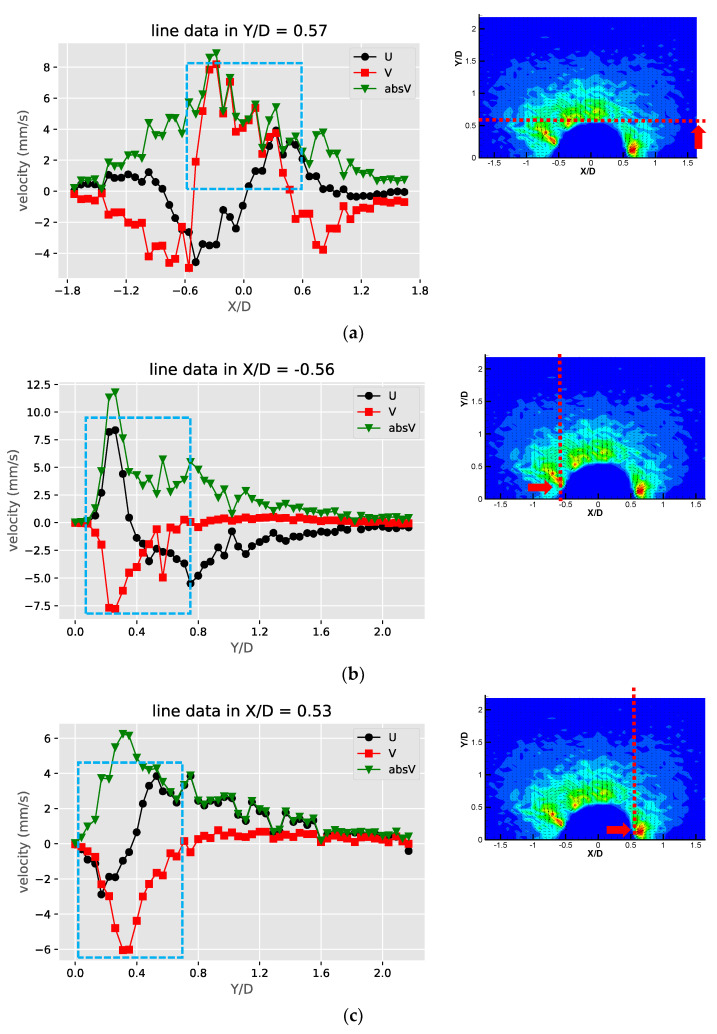
The profiles of velocity components in the X and Y directions (U, V) and absolute velocity (absV) of the liquid droplet obstructions: (**a**) Y/D = 0.57; (**b**) X/D = −0.56; (**c**) X/D = 0.53 and (**d**) X/D = −0.01.

**Figure 8 micromachines-11-00891-f008:**
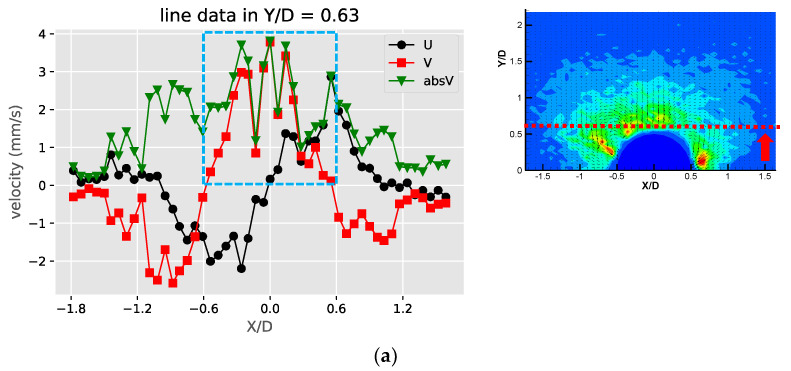
The profiles of velocity components in the X and Y directions (U, V) and absolute velocity (absV) of the gaseous bubble obstructions: (**a**) Y/D = 0.63; (**b**) X/D = −0.47; (**c**) X/D = 0.48 and (**d**) X/D = 0.0.

**Figure 9 micromachines-11-00891-f009:**
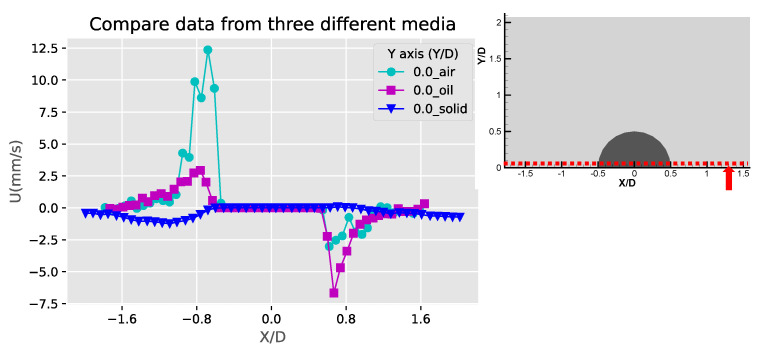
Comparison of the X-component velocity (U) profiles at the base of the obstructions (Y/D = 0).

**Figure 10 micromachines-11-00891-f010:**
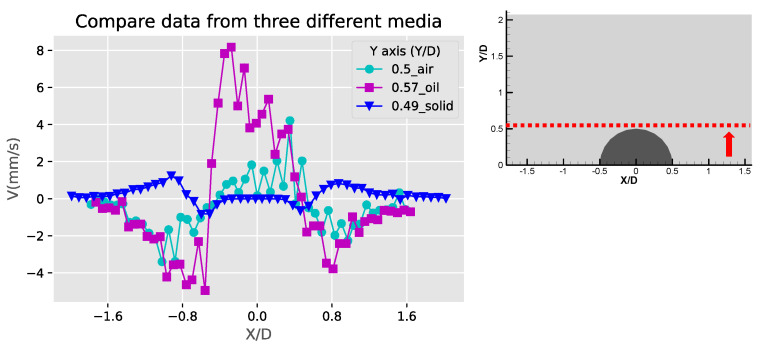
Comparison of the Y-component velocity (V) profiles at the top of the obstructions (Y/D = 0.5).

**Figure 11 micromachines-11-00891-f011:**
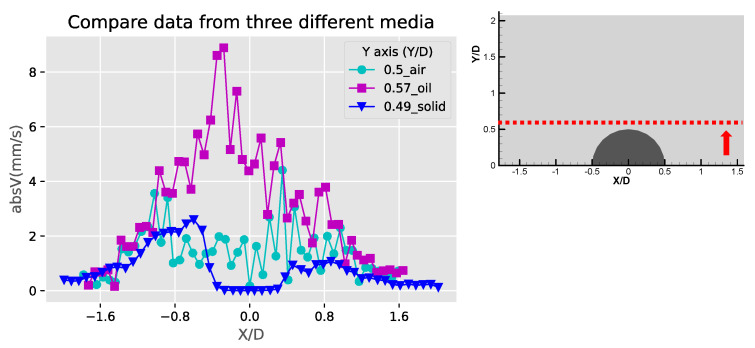
Comparison of the absolute velocity (absV) profiles at the top of the obstructions (Y/D ≅ 0.6).

**Figure 12 micromachines-11-00891-f012:**
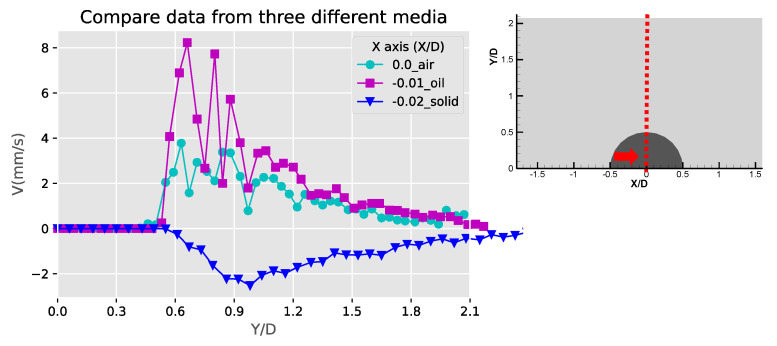
Comparison of the Y-component velocity (V) profiles at the centerline of the obstructions (X/D = 0).

**Figure 13 micromachines-11-00891-f013:**
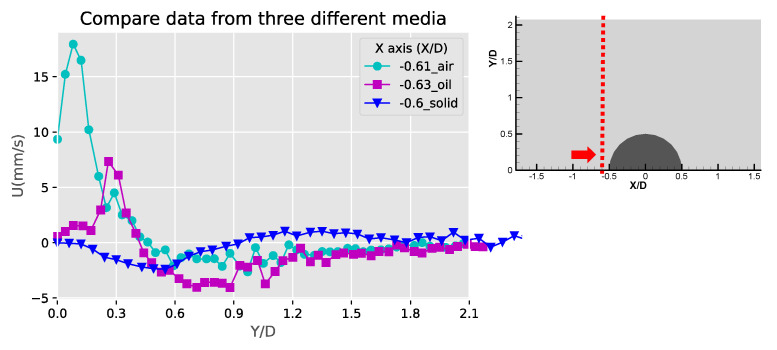
Comparison of the X-component velocity (U) profiles at the left side of the obstructions (X/D = −0.6).

**Figure 14 micromachines-11-00891-f014:**
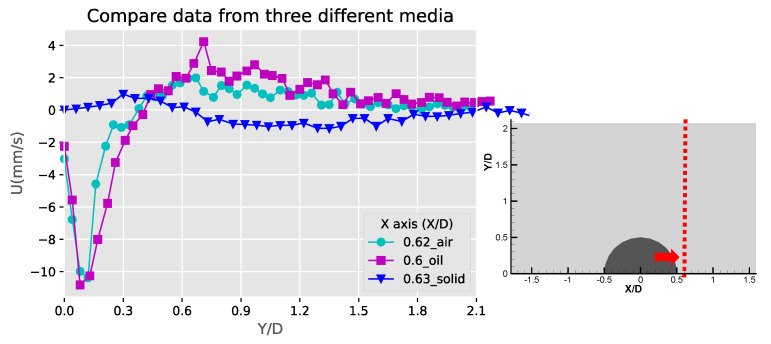
Comparison of the X-component velocity (U) profiles at the right side of the obstructions (X/D = 0.6).

**Figure 15 micromachines-11-00891-f015:**
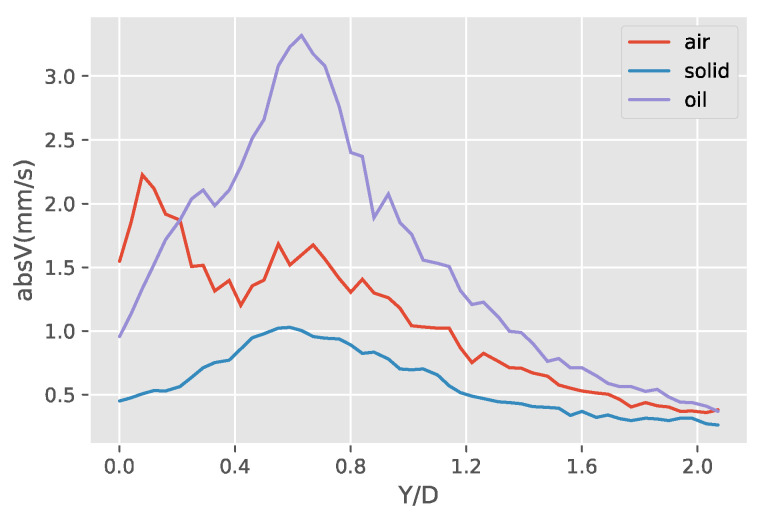
The absolute velocity profile averaged over the *X*-axis along the *Y*-axis of the three obstructions.

**Figure 16 micromachines-11-00891-f016:**
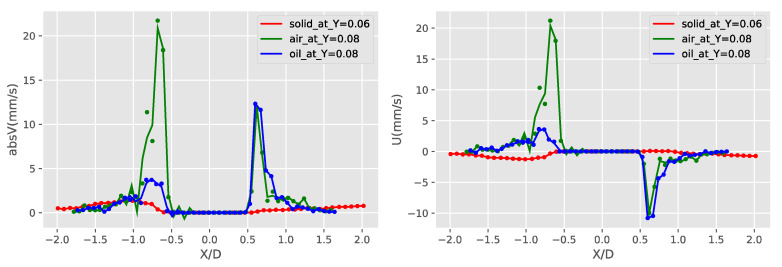
Comparison of the absolute velocities (absV) and X-component velocity (U) of the three types of obstruction Y/D = 0.08 (location of the gaseous bubble bubble velocity peak).

**Figure 17 micromachines-11-00891-f017:**
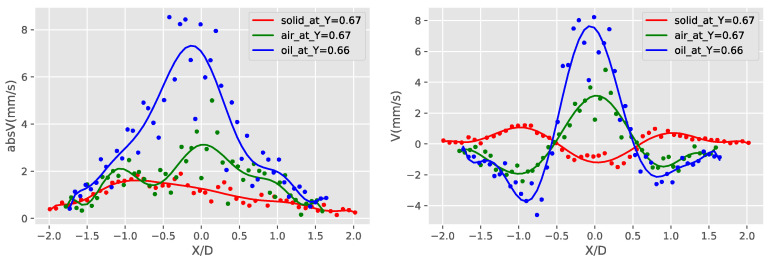
Comparison of the absolute velocities (absV) and Y-component velocity (V) of the three types of obstruction Y/D = 0.66 (location of the liquid droplet velocity peak).

**Figure 18 micromachines-11-00891-f018:**
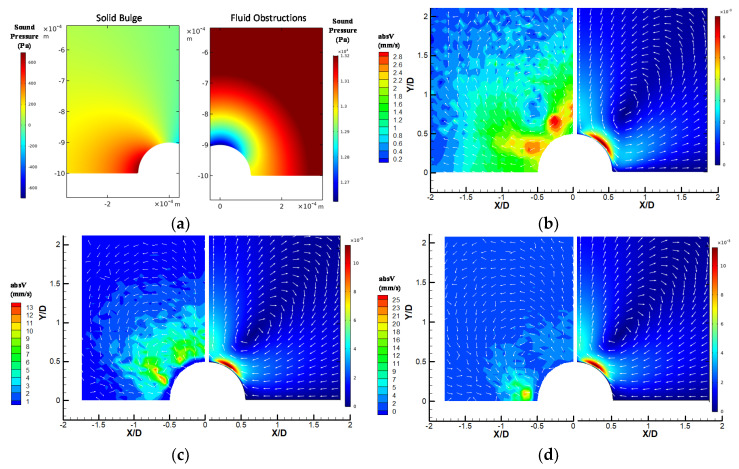
Numerical simulation results and comparisons to experimental results: (**a**) acoustic pressure distribution of the solid bulge (left) and fluid obstruction case (right); (**b**–**d**) side-by-side comparisons of distributions of streaming velocity vector and absolute velocities (absV) of (**b**) solid, (**c**) liquid droplet and (**d**) gas bubble obstruction. Experimental results are on the right side and numerical simulations are on the left side in each subplot.

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
