# Peer review of "Comparison of Acoustic Streaming Flow Patterns Induced by Solid, Liquid and Gas Obstructions"

_micromachines, 2020, doi:10.3390/mi11100891_

Round 1
Reviewer 1 Report
Lu et al. studied the acoustic streaming patterns generated with different obstructions. While acoustic streaming around gas bubbles and PDMS structures have been reported before, the study on the streaming patterns at the interface of two-phase liquid is new. I recommend the publication of this work after addressing the following concerns:
- More discussion should be given on the comparison of different obstructions. What insights can be drawn from the liquid obstruction results? Why does the liquid obstruction generate higher streaming velocity than the PDMS structure?
- During the experiment, are the liquid and gas bubbles stable over the time? Any change in shape or size? If there is any change, does it affect the results?
- More explanation should be given on the claim “liquid droplet transfers the oscillation energy most efficiently”.
- A schematic should be provided to show the relative position between the transducers and the microchannel.
- There is a typo in the last sentence of the abstract.
Reviewer 2 Report
The authors present an experimental study of streaming in the presence of solid, liquid and gaseous obstructions. Since two of the three forms of obstruction have already been studied, this manuscript describes a small incremental step. Critically, the manuscript lacks support from predictions or models of any type. This means it would be nearly impossible for a reader to make use of these results in any other but the author's experimental configurations. For this reason, this manuscript is not ready for publication. Additional comments follow:
L88. Make clear the novelty and significance of the study.
L94. Does the manual control of droplet / gas volume induce any experimental errors? What could be done to mitigate this issue?
L97. Explain why the listed dimensions were chosen
L127. What is the distribution of acoustic pressure in the channel under the specified drive conditions? This information provides critical context for the streaming results. This is also relevant to the author's comment at L204.
L167. Are the plotted results in this section for a single experiment? Were the results confirmed through performing multiple experiments?
L192. In Figure 4, list the units of X and Y
L269. The discussion section needs to have a focused description of study limitations and what could be done to improve upon them in the future.
Typographical errors were found at L22, L50, L74, L78, and many other places. The authors should carefully review the text before resubmitting the manuscript.
Round 2
Reviewer 1 Report
My previous comment "A schematic should be provided to show the relative position between the transducers and the microchannel." has not been addressed.
Reviewer 2 Report
The authors made substantial changes to improve the manuscript, and the effort is appreciated. The addition of simulation results is quite helpful, but more detail is needed as below.
1. Emphasize both in the abstract and conclusions that the experiment results are supported by numerical simulations.
2. The modeling described in L371-L384 should be moved to the methods section, and more detail must be provided so that readers may fully understand and attempt to replicate the results. Any prior work done to validate the simulation approach should clearly be described.
3. The simulations would be most impactful if placed side by side with corresponding measurements. Otherwise it is difficult for a reader to appreciate the similarities.
L82: Has this comparison been done in simulation for all three types?
L104: Provide a value for the available precision.
L117: Are there results from the authors' model or others that can quantify the impact of obstruction radius on the resulting fields? What are the other uncertainties and errors in the experiments. These need to be quantified if the authors want to support the claim that obstruction volume change effects are negligible.
L125: Fig 2 is a bit difficult to interpret relative to the full experiment. Please expand or provide more detail relating it to Fig 3.
L405-410: This very long passage is challenging to follow. Please split and clarify.
L411-414: The authors should use their model to explore at least one supporting example - with a valid simulation capability, there is no need to speculate on the role of liquid/gas properties.
L428 and elsewhere: discussion of kinetic energy and its transfer appears often in this manuscript, but the actual energies were not quantified. Are the authors able to confirm through their simulations that the efficiency of energy transfer is as they say?
Fig 18 legend says (b) instead of (d) for the fourth panel.
Round 3
Reviewer 2 Report
The authors have substantially revised the manuscript in line with reviewer comments. Three recommendations remain:
1. In Figs 16 and 17, identify what the side by side plots are, and show them on identical axis scales.
2. Add a discussion of why there are clear qualitative differences between the model and simulation. For example, the simulations show the largest near-obstruction velocities near 45 degrees, where the measurements do not. Also, it is difficult to see the streaming patterns in experiments (b) and (c). Can they be re-scaled for easier comparison between cases and the simulations?
3. Perform another review of the manuscript for grammar and typographical errors.
